# The Structural Flexibility of PR-10 Food Allergens

**DOI:** 10.3390/ijms23158252

**Published:** 2022-07-26

**Authors:** Sebastian Führer, Jana Unterhauser, Ricarda Zeindl, Reiner Eidelpes, Monica L. Fernández-Quintero, Klaus R. Liedl, Martin Tollinger

**Affiliations:** 1Institute of Organic Chemistry and Center for Molecular Biosciences Innsbruck (CMBI), University of Innsbruck, Innrain 80/82, 6020 Innsbruck, Austria; bastif92@web.de (S.F.); jana.unterhauser@uibk.ac.at (J.U.); ricarda.zeindl@uibk.ac.at (R.Z.); reiner.eidelpes@ki.se (R.E.); 2Department of General, Inorganic and Theoretical Chemistry and Center for Molecular Biosciences Innsbruck (CMBI), University of Innsbruck, Innrain 80/82, 6020 Innsbruck, Austria; monica.fernandez-quintero@uibk.ac.at (M.L.F.-Q.); klaus.liedl@uibk.ac.at (K.R.L.)

**Keywords:** protein flexibility, conformational breathing, ligand binding, NMR spectroscopy, relaxation dispersion

## Abstract

PR-10 proteins constitute a major cause of food allergic reactions. Birch-pollen-related food allergies are triggered by the immunologic cross-reactivity of IgE antibodies with structurally homologous PR-10 proteins that are present in birch pollen and various food sources. While the three-dimensional structures of PR-10 food allergens have been characterized in detail, only a few experimental studies have addressed the structural flexibility of these proteins. In this study, we analyze the millisecond-timescale structural flexibility of thirteen PR-10 proteins from prevalent plant food sources by NMR relaxation-dispersion spectroscopy, in a comparative manner. We show that all the allergens in this study have inherently flexible protein backbones in solution, yet the extent of the structural flexibility appears to be strikingly protein-specific (but not food-source-specific). Above-average flexibility is present in the two short helices, α1 and α2, which form a V-shaped support for the long C-terminal helix α3, and shape the internal ligand-binding cavity, which is characteristic for PR-10 proteins. An in-depth analysis of the NMR relaxation-dispersion data for the PR-10 allergen from peanut reveals the presence of at least two subglobal conformational transitions on the millisecond timescale, which may be related to the release of bound low-molecular-weight ligands from the internal cavity.

## 1. Introduction

Pathogenesis-related (PR) proteins represent a major source of allergic reactions to plants, and they play a major role in the defense mechanisms to external stimuli or pathogenic infections [1]. In the last decades, a vast number of PR proteins have been discovered, studied, and classified into different groups based on their similarity regarding their biological activities, sequences, and regulatory or physicochemical properties. Currently, PR proteins are divided into 17 subfamilies, with distinct biological features [2,3]. Highly prevalent and important is the family of PR proteins in class 10 (PR-10), which were first discovered in parsley [4], and which comprise hundreds of members across the plant kingdom. Their role in the general defense mechanism of plants is evident by the fact that these proteins are upregulated after viral, bacterial, or fungal invasion, or due to abiotic factors, such as cold, aridity, oxidative stress, and UV radiation, or, in general, by phytohormones [5].

The eponymous and most prominent PR-10 allergen is the birch pollen (*Betula verrucosa*) protein Bet v 1, which was first characterized in 1983 [6], and which provokes immediate hypersensitivity reactions of type I, resulting in the development of Bet v 1-specific IgE antibodies. The so-called cross-reactivity to other food and pollen allergens has been observed in clinical studies [7,8,9], and it arises from the highly similar amino acid sequences and three-dimensional structures of these proteins. This leads to the presence of analogous and overlapping epitopes in these proteins, which are recognized by IgE antibodies, which trigger an allergic response [10]. Allergic symptoms include the oral allergy syndrome, which is characterized by the itching and swelling of the throat, tongue, and lips [11], along with rare anaphylactic shocks in birch-pollen-sensitized individuals [12,13]. The most prevalent Bet v 1 cross-reactive plant food sources are apple (inducing allergic symptoms in 80% of Bet v 1-sensitized patients), hazelnut (59%), peach (51%), kiwi (48%), peanut (24%), and plum (24%) [14].

PR-10 allergens are of similar molecular sizes (156 to 160 residues) and masses (~17.5 kDa). A striking feature of PR-10 allergens is their highly conserved and characteristic secondary structures, which comprise three α-helices, along with a seven-stranded antiparallel β-sheet (Figure 1). The first strand, β1, is connected to strand β2 via two short helices, α1 and α2. Together with β1 and β2, the remaining five β-strands (β3–β7) complete the β-sheet, which displays a pronounced curvature. The structural composition is closed by a long C-terminal helix (α3), which is located above the β-sheet and the two short helices α1 and α2. Generally, PR-10 allergens display medium to high amino acid-sequence identities among each other, with different levels of conservation along the sequence. While the glycine-rich region (GXGGXG) in the loop connecting strands β2 and β3 (positions 46–51, using the numbering scheme of the birch-pollen allergen Bet v 1) is highly conserved in PR-10 allergens, the most variable region regarding both length and sequence is located at the C-terminus (Supporting Appendix A) [1,5].

The tertiary fold of PR-10 proteins comprises a large internal cavity of a vastly variable size, with volumes up to ~4000 Å^3^ [5]. While the cavity is, in large parts, formed by hydrophobic amino acids, several hydrophilic amino acids with hydrogen-bond donor/acceptor capacities confer a certain degree of amphiphilicity to its surface [15]. Entrances to the cavity are located between helix α3 and loops L3, L5, and L7 (ε1, the largest opening), and in some PR-10 proteins, between helix α3 and strand β1 (ε2) [5,16]. PR-10 proteins promiscuously bind a vast variety of low-molecular-weight ligands to their internal cavities with different affinities [5], including fatty acids, flavonoids, phenolic acids, cytokinins, and steroids [17,18]. Whether and how the ligand binding to PR-10 allergens is related to their immunologic properties are currently under debate [19,20].

The biophysical research on PR-10 allergens in the last decades has focused on ligand binding and structure elucidation by NMR spectroscopy or X-ray crystallography, and on understanding how the structural properties of these proteins are related to the observed allergenic behavior [16]. Only a few studies have addressed the structural flexibility of these proteins, which is a feature that may be related to the affinity of low-molecular-weight ligands binding to the internal cavity, and IgE binding to the outer surfaces of these proteins [21,22,23,24]. In this work, we provide a thorough and quantitative comparison of the structural flexibilities of 12 PR-10 food allergens from apple, hazelnut, peach, kiwi, peanut, and plum, along with—and in comparison to—the sensitizing birch-pollen allergen, as observed by solution NMR spectroscopy. The possible implications for low-molecular-weight ligand binding and interactions with antibodies are discussed.

## 2. Results

In order to generate a comprehensive dataset covering the most important food sources, we chose thirteen different PR-10 allergens (and the isoforms thereof): Pru p 1 from peach (*Prunus persica*), Cor a 1 from hazelnut (*Corylus avellana*), Mal d 1 from apple (*Malus domestica*), Ara h 8 from peanut (*Arachis hypogaea*), Act c 8 from golden kiwi fruit (*Actinidia chinensis*), Act d 8 from green kiwi fruit (*Actinidia deliciosa*), and Pru d 1 from plum (*Prunus domestica*), along with the sensitizing allergen from the pollen of birch trees (*Betula verrucosa*), Bet v 1.

### 2.1. Surface Properties of the PR-10 Allergens

Figure 2 compares the surface properties of these proteins. The clustering based on the charge profiles and H-bond donor/acceptor capacities of the surfaces, identified and classified by the YRB highlighting scheme of Hagemans et al. [25], shows that these proteins differ significantly regarding their surface properties, which is in agreement with their considerable sequence variability. Cluster 1 comprises the isoforms Cor a 1.0401, Cor a 1.0404, Mal d 1.0101, and Pru p 1.0101, while Cluster 2 includes Ara h 8.0101, Act d 8.0101, Bet v 1.0101, Cor a 1.0402, Cor a 1.0403, Pru d 1.0101, Pru p 1.0201, and Pru p 1.0301. Cluster 3 only contains Act c 8.0101. On the one hand, the three clusters differ particularly regarding the distribution of charged residues, which are mostly present in the first cluster. Cluster 2, on the other hand, contains a higher number of hydrophobic residues and less charged residues on the surface of the helix α3. The golden kiwifruit allergen Act c 8.0101 forms a separate cluster, which can be explained by the high number of surface-exposed cysteine residues (Act c 8.0101 contains seven cysteines, with six of them located on the protein surface). Moreover, all three clusters show high densities of amino acid residues, with the hydrogen-bond capacity on the surface. This comparison emphasizes that these PR-10 allergens are remarkably diverse with regard to the electrostatic properties of the surface that is exposed to the solvent, which is in line with the observation that their immunologic properties, including their cross-reactivity and IgE binding, are diverse [26,27,28], despite the fact that these allergens display highly similar three-dimensional scaffolds (Supporting Appendix A).

### 2.2. Millisecond-Timescale Flexibility of PR-10 Allergens

We used backbone amide ^15^N relaxation-dispersion (RD) NMR experiments to probe the structural flexibility of the selected PR-10 allergens by experiment. RD data contain quantitative information about the structural flexibility that occurs on the millisecond timescale on a per-residue basis [29]. As noted previously for PR-10 allergens from birch pollen and hazelnut, these proteins are remarkably flexible, with a significant portion of the backbone amides showing nonflat relaxation-dispersion curves [21,22]. Figure 3A compares the backbone amide ^15^N relaxation-dispersion amplitudes (ΔR_2,eff_ values) of the thirteen PR-10 proteins in this study, in detail. Significant differences regarding the percentages of amino acid residues with nonflat RD profiles (ΔR_2,eff_ exceeding 1 s^−1^) are evident, ranging from 50% for the plum allergen Pru d 1, to more than 80% in different isoforms of the peach and hazelnut allergens, Pru p 1 and Cor a 1, respectively. Interestingly, however, these values do not correlate with the amino acid sequences in a straightforward manner. For example, peach Pru p 1.0101 and plum Pru d 1.0101, one of the most disparate pairs of allergens regarding the percentage of flexible residues, share 99.4% of their sequence identity and have only one amino acid substituted, at position 117 in the center of strand β7. Cor a 1.0401 and Cor a 1.0404, which are two isoforms of the hazelnut PR-10 allergen, share 98.8% of their sequence identity, but they appear to be significantly different regarding the extent of the structural flexibility in their protein backbones. Indeed, these data reinforce the notion that different isoforms from one particular food source (e.g., hazelnut) can have substantially different backbone flexibilities, even if the sequence identities are high (>96% for the four hazelnut PR-10 isoforms in this study), while, for other food sources (e.g., peach), isoforms with much lower sequence identities (<80%) can have similar backbone flexibilities. Moreover, Figure 3A indicates that the birch pollen Bet v 1.0101, peach Pru p 1.0301, peanut Ara h 8.0101, and golden kiwi Act c 8.0101 are the PR-10 proteins with the highest percentages of residues with ΔR_2,eff_, exceeding 10 s^−1^, while apple Mal d 1.0101, hazelnut Cor a 1.0404, and green kiwi Act d 8.0101 display particularly small numbers of amino acids, with ΔR_2,eff_ ≥ 10 s^−1^. To conclude, it is evident from this compilation that, while all PR-10 allergens investigated in this study are structurally flexible, the extent of the structural flexibility is strikingly protein-specific.

Figure 3B,C compile the available experimental information regarding the location of the structural flexibility in these proteins. Evidently, the structural flexibility is distributed along the entire backbone of all thirteen proteins, rather than being limited to the loops or turns of the PR-10 fold. Surprisingly high levels of flexibility are present in all secondary-structure elements: α1–α3 and β1–β7. A region with above-average structural flexibility is situated between residues 20 and 40, which corresponds to the two short helices α1 and α2 that form a V-shaped support for the long C-terminal helix α3, which itself also contains flexible amino acid residues (in particular, at its center, between residues 135–148). As a matter of fact, this holds true for all the PR-10 proteins in this study (the relaxation-dispersion amplitudes for individual PR-10 proteins are shown in Supporting Appendix A). Nonflat RD profiles are also found for the β-sheet, and especially at the beginning of strand β2, which packs to helix α2 at the tip of β3, which is directly adjacent and hydrogen-bonded to strand β2, in the center of strand β7. In addition, surface-exposed loops between these secondary-structure elements (e.g., loop L2 connecting helices α1 and α2, and loop L3 connecting helix α2 with strand β3) display considerable millisecond-timescale flexibility. For loop L5, which is located between strands β3 and β4, the NMR resonances are broadened beyond detection in the majority of the PR-10 proteins, which indicates high levels of structural flexibility.

Together, some of these flexible segments (helices α1, α2, strands β2 and β3, and the central part of helix α3) shape the surface of the inner cavity of these proteins, which suggests that the cavity itself is malleable. In addition, a number of amino acid residues that form the largest entrance (ε1) to the cavity (loops L3 and L5, helix α3) display above-average flexibility. This is clearly contrasted by the glycine-rich loop (L4, connecting strands β2 and β3) and loop L6 (between β4 and β5), which display lower percentages of residues with nonflat relaxation-dispersion profiles.

### 2.3. Millisecond-Timescale Flexibility of Ara h 8

We chose the peanut allergen Ara h 8.0101 for an in-depth analysis of the backbone millisecond flexibility and ligand binding to the inner cavity. For this protein, which is a minor allergen that is present in peanut that shows IgE cross-reactivity with Bet v 1.0101, a high-resolution X-ray structure in a complex with (-)-epicatechin is available [30]. NMR resonance assignments for Ara h 8.0101 were obtained using triple-resonance experiments, as described [31], and the backbone amide ^1^H-^15^N-HSQC spectrum of this allergen is shown in Figure 4A. Excluding the N-terminal glycine residue, we were able to assign the backbone amide resonances of 144 (out of 146) nonproline residues of Ara h 8.0101, corresponding to 98.6% completeness. The majority of the resonances are well dispersed in the ^1^H-^15^N-HSQC spectrum, and the backbone amide ^15^N relaxation-dispersion data are accessible for 95.1% of all the backbone amide resonances. A total of 69.2% of the backbone amides in Ara h 8.0101 show nonflat relaxation-dispersion profiles, with RD amplitudes exceeding 1 s^−1^ at 700 MHz, which is the median of the allergens in this study. This particular protein thus constitutes a representative example of PR-10 food allergens.

We analyzed the backbone amide ^15^N relaxation-dispersion data of Ara h 8.0101, recorded at 600 MHz and 700 MHz, by fitting the equation derived by Baldwin [32] to the experimental data. Initially, we attempted to fit a single global conformational exchange process (transition) between two states to the experimental data for all the residues by assuming a common exchange-rate constant (k_ex_), along with uniform populations, but allowing for residue-specific values of the chemical-shift difference between the two states (Δδ_ex_), as described in the Methods section. This resulted in fits of insufficient quality and low-convergence performance (data not shown). However, the fitting procedure improved significantly, when we assumed that two distinct processes were present, by grouping amino acid residues in Ara h 8.0101 according to their positions in the secondary-structure elements (Figure 4B). One subglobal cluster contained all the residues that had nonflat relaxation-dispersion profiles in helices α1, α2 (including the short loop L2 between α1 and α2), and α3 (20 residues in total), while the second subglobal cluster comprised all the other residues, located in the central β-sheet and in loops L1 or L3–L9 (26 residues). For each cluster, we again assumed uniform exchange-rate constants and populations, but residue-specific Δδ_ex_ values. Figure 4C,D shows the exemplary relaxation-dispersion data and fitting curves for these two groups of residues. It is evident from a visual inspection of the data for these amino acids that they are dissimilar regarding the shapes of the RD profiles. The residues in the β-sheet and in the loops (Lys37, Ser38, Thr55, Ile56, Leu66, Asp73, Val84, Ala88 in Figure 4C) have rising profiles with flat plateaus at low CPMG frequencies, whereas the three α-helices (Ala21, Ser27, Lys31, Glu128, Gly134, Lys137, Glu139, Glu146 in Figure 4D) display ascending RD profiles. This suggests that the underlying conformational exchange processes are not identical.

The fitting of the experimental data shows that the two processes are indeed distinct: the exchange rate for the subglobal conformational transition comprising helices α1–α3 is k_ex_ = 1060 ± 70 s^−1^, with a minor state population (7.9 ± 1.4%), while a somewhat slower transition with a lower minor state population is present in the β-sheet (k_ex_ = 810 ± 80 s^−1^, minor state population: 1.3 ± 0.1%). These figures translate to slightly different timescales for the β-sheet (1.2 ± 0.1 ms) and the three helices (0.9 ± 0.1 ms). Of note, the residue-specific ^15^N chemical-shift differences that were obtained from the fits of the relaxation-dispersion data (Δδ_ex_) are quite substantial (Appendix A). For the backbone amides in the β-sheet, Δδ_ex_ values exceeding 3 ppm (^15^N) were found for residues in strands β2 (Ser38), β4 (Leu66, His67), β5 (Val84), β6 (Thr97, Phe98), and loop L7 (Ala88). Such large Δδ_ex_ values indicate the transient loosening or even unfolding of the β-sheet. Hydrogen/deuterium exchange experiments showed that the hydrogen bonds in the four hazelnut Cor a 1.040x isoforms are indeed labile, and amide protons are susceptible to exchange with the surrounding water [21]. For helices α1–α3, the Δδ_ex_ values are generally smaller (<2.5 ppm), and they exceed 1.5 ppm only for Ala21, Ser27, and Lys31 in the two helices (α1 and α2) that form the V-shaped support for α3. A comparison of the Δδ_ex_ values with the chemical-shift differences that are expected for the unfolding of the protein backbone shows that these values are not correlated for the β-sheet and the three helices (Appendix A). Rather, the experimental Δδ_ex_ values are systematically smaller than the unfolding shift changes. This implies that the conformational transitions that we observe by relaxation-dispersion NMR do not result from the unfolding of the protein scaffold to a random coil. Nevertheless, the considerable size of the observed chemical-shift differences indicates that the underlying conformational transitions involve the local restructuring and/or loosening of the hydrogen bonds in the secondary structures.

### 2.4. Ligand Binding to Ara h 8

We probed the interaction of (-)-epicatechin with Ara h 8.0101 in solution in an NMR titration experiment. Hurlburt et al. showed that this naturally occurring flavan-3-ol binds to an amphiphilic pocket inside the cavity of Ara h 8, which is formed by helices α1–α3 and strands β2–β5 (Figure 5) [30]. The addition of (-)-epicatechin to Ara h 8.0101 leads to a gradual change in the chemical shifts of the resonances, in accordance with an intermediate-to-fast binding process on the NMR chemical-shift timescale, as is typically observed for low-affinity binding (Figure 5A). The largest chemical-shift changes of the backbone ^1^H-^15^N-resonances (Δδ_obs_) are found for residues in helices α1 (Met22) and α2 (Ala25 and Lys31), loop L3 (Ile33 Asp34, and Val36), strand β4 (Thr55, Ile56, and Val57), loop L7 (Val87), strand β6 (Phe98), and for the central part of the C-terminal helix α3 (Gly138, Leu141, and Ile145). These amino acid residues cluster around the (-)-epicatechin binding site in the crystal structure (Figure 5C). Indeed, in this complex, similar amino acid residues (Asp26, Thr29, Ile56, His67, Tyr81, Lys137, and Leu141) were identified as the primary interaction sites of (-)-epicatechin via hydrogen bonding and noncovalent interactions [30], which confirms that, in solution, this ligand binds to Ara h 8.0101 in a similar manner as in the crystalline state. In the quantitative analysis of the NMR titration data for backbone amide resonances with chemical-shift perturbations (Δδ_obs_), exceeding 0.1 ppm yields an average complex dissociation constant (K_D_) of 0.76 ± 0.37 mM. Interestingly, 53.3% of these backbone amides exhibit relaxation-dispersion amplitudes exceeding 5 s^−1^ in the apo-form of the protein, while only 20.3% of all the other backbone amides in this protein fall into this category. Likewise, 20.0% of the interacting residues have RD amplitudes exceeding 10 s^−1^, contrasting only 7.6% in the remainder of the backbone. This suggests that the (-)-epicatechin binding site displays above-average flexibility before the ligand is bound.

We next employed ligand-observed relaxation-dispersion experiments to investigate the kinetics of the (-)-epicatechin binding to Ara h 8.0101 [34]. The two aromatic protons (^1^H-6 and ^1^H-8) of (-)-epicatechin (Figure 5B) were resolved in 600 MHz ^1^H-NMR spectra, with a scalar ^4^J_HH_ coupling constant of <2 Hz, which qualified them for ^1^H-CPMG relaxation-dispersion experiments. Figure 5D shows the experimental ^1^H-CPMG RD profiles recorded for an increasing saturation (0–12%) of the ligand with Ara h 8. The RD profile for (-)-epicatechin in the absence of Ara h 8 is perfectly flat, which indicates that scalar coupling does not interfere with the experiment. The addition of protein produces nonflat RD profiles, with a clear dependence on the saturation level, which is indicative of the ligand binding and release in the millisecond time regime. The fitting of the data with a two-site kinetic model yields the rate constants that describe the ligand release for ^1^H-6 (k_off_ = 1600 ± 300 s^−1^) and ^1^H-8 (k_off_ = 1300 ± 300 s^−1^), which translates to a residence time of (-)-epicatechin in the protein-bound state of just below one millisecond (0.6 ± 0.1 ms and 0.8 ± 0.2 ms, respectively). 

## 3. Discussion

The accumulated NMR spectroscopic data in this study show that, in solution, PR-10 allergens are remarkably flexible. In some of these proteins, more than 80% of all the amino acid residues have nonflat relaxation-dispersion curves, which indicates the backbone flexibility that occurs on the millisecond timescale. Quite generally, the most flexible segments of PR-10 proteins are centered around helices α1 and α2, which form the V-shaped support for the long C-terminal helix α3. Together with the curved β-sheet, these helices shape the internal binding cavity of these proteins, which suggests that the surface of the cavity itself is structurally heterogeneous and is capable of adjusting to different ligands [35]. Conformational variability in the internal cavity has been observed in the crystal structures of various PR-10 proteins, such as the strawberry allergen Fra a 1, where the C-terminal helix α3 bends toward the central β-sheet when the ligand is bound, forming a more compact and closed three-dimensional structure [36]. Likewise, loop L5, at the entrance to the internal cavity of this allergen, appears to be conformationally heterogenous in the crystalline state, but it adopts a more closed conformation upon ligand binding [37]. In solution, most NMR resonances in loop L5 of PR-10 food allergens are broadened beyond detection due to conformational heterogeneity.

It is possible that the inherent flexibility of PR-10 proteins represents a structural reorganization process that is required for ligand entry and/or release. An in-depth analysis of the NMR experimental data of the peanut allergen Ara h 8 suggests that structural flexibility is reminiscent of “conformational breathing”. The chemical-shift data indicate that the structural changes that arise from this process are quite substantial and might involve the loosening of hydrogen bonds, even in secondary-structure elements. This is in accordance with our previous observation that structural flexibility in PR-10 proteins enhances the exposure of the protein backbone to the surrounding water [21]. The conformational breathing of the PR-10 scaffold appears to involve at least two subglobal processes, both occurring on the millisecond timescale (1.2 ± 0.1 ms for the β-sheet, and 0.9 ± 0.1 ms for the three α-helices). These values qualitatively agree with the residence time of (-)-epicatechin in the bound state, which was found to be just below one millisecond (0.6–0.8 ms). While these values suggest that ligand release may indeed be mechanistically coupled to the inherent structural flexibility of the PR-10 scaffold, additional studies using a comprehensive set of ligand molecules with diverse affinities will be required to corroborate this observation.

Structural flexibility also plays an important role regarding the interactions between allergens and antibodies [38]. Antibody binding may be accompanied by rigidification and a loss of conformational entropy, involving both the antigen’s paratope, as well as the epitope on the surface of a particular allergen. For the four hazelnut Cor a 1.040x isoforms, an inverse correlation between the overall structural flexibility of the allergen and IgE binding has indeed been recently reported [21]. It could be shown that more flexible isoforms of this particular allergen have lower IgE binding potential than more rigid ones. The compilation of the data presented here clearly illustrates that the structural flexibility in PR-10 allergens is not evenly distributed along the protein backbone. The glycine-rich loop, which has been identified as a cross-reactive IgE epitope in some of these allergens, is particularly rigid and highly conserved, and it has a conformation that is compatible with antibody binding [15,35,39]. This is contrasted by the C-terminal helix α3, in which the conformational flexibility is pronounced, and the variation in the primary sequence is high. Additionally, helix α3 has been identified as an important IgE epitope in Bet v 1 [40].

These examples suggest that entropic contributions that arise from epitope rigidification may well be present in some—but probably not in all—cases of antibody/antigen binding. A combination of the (static) structural data and site-specific information about the structural flexibility are required to fully comprehend these interactions.

## 4. Materials and Methods

### 4.1. Recombinant Protein Expression and Purification

The construction of plasmids encoding for the PR-10 proteins, and the subsequent expression and purification, has been described for Act c 8.0101, Act d 8.0101 [41], Bet v 1.0101 [23], Cor a 1.0401, Cor a 1.0402, Cor a 1.0403, Cor a 1.0404 [31], Mal d 1.0101 [42], and Pru p 1.0101 [43]. The protocols for Ara h 8.0101 (accession no. AAQ91847), Pru d 1.0101 (accession no. ABW99634), Pru p 1.0201 (accession no. AJE61290), and Pru p 1.0301 (accession no. AJE61291) were adapted from [43]. Amino acid sequences are provided in the Supporting Appendix A. All NMR samples contained 20 mM sodium phosphate buffer (pH: 6.9), 9% D_2_O, and a varying amount of ^15^N-labeled protein (0.2–0.8 mM). Only Bet v 1.0101 was dissolved in 5 mM sodium phosphate buffer (pH: 8.0) (9% D_2_O). The Cor a 1.040x samples, in addition, contained 2 mM dithiothreitol (DTT).

### 4.2. NMR Relaxation-Dispersion Experiments

Backbone amide ^15^N relaxation-dispersion experiments for each 0.5 mM protein sample were recorded at 25 °C on a 700 MHz Bruker Avance Neo spectrometer equipped with a Prodigy CryoProbe, using sensitivity-enhanced Carr–Purcell–Meiboom–Gill (CPMG) sequences [44], with the continuous-wave decoupling of ^1^H during the CPMG period [45]. Different CPMG field strengths (ν_CPMG_) of 33.3, 66.7, 100.0, 133.3, 166.7, 200.0, 266.7, 333.3, 466.7, 600.0, 733.3, and 933.3 Hz were applied, with repeat experiments at 66.7 Hz and 600.0 Hz. The length of the CPMG pulse train was set to T_relax_ = 30 ms. Partial peak volumes (intensities in 5 × 5 grids) were analyzed with nmrDraw [46] to determine the effective relaxation rates: R_2,eff_ = –1/T_relax_·ln(I/I_0_), with I being the intensity at a given ν_CPMG_, and I_0_ being the reference intensity with the T_relax_ set to 0. Uncertainties in relaxation rates (σ) were calculated from the repeat experiments. The obtained relaxation-dispersion profiles were fitted with an in-house MATLAB script by using the equation derived by Baldwin [32] to extract the per-residue relaxation-dispersion amplitudes (∆R_2,eff_) as ∆R_2,eff_ = R_2,eff_ (ν_CPMG_ = 0) − R_2,eff_ (ν_CPMG_ = ∞). Here, R_2,eff_ (ν_CPMG_ = ∞) and R_2,eff_ (ν_CPMG_ = 0) refer to the extrapolated R_2,eff_ values at infinite and zero ν_CPMG_ field strengths, respectively.

For Ara h 8.0101, the 700 MHz relaxation-dispersion data were complemented by experiments recorded at 600 MHz, using a Bruker Avance II+ spectrometer equipped with a Prodigy CryoProbe. The simultaneous fitting of the data at the two magnetic field strengths was performed. The 600 MHz and 700 MHz relaxation-dispersion profiles were fitted to global two-site exchange models, assuming uniform exchange-rate constants (k_ex_) for the transition between two states (A and B), and uniform populations of states A and B (p_A_ and p_B_, with p_A_ = 1 − p_B_) for all residues in the fit, but residue-specific values of |Δδ_ex_|. In fits assuming the presence of two distinct exchange processes, residues with relaxation-dispersion amplitudes (∆R_2,eff_) > 1 s^−1^ at 700 MHz were grouped into those belonging to β-strands and loops (residues Lys37, Ser38, Val39, Ile41, Glu43, Gly47, Thr50, Lys53, Thr55, Ile56, Leu66, His67, Asp73, Glu74, Val84, Val87, Ala88, Leu89, Thr97, Phe98, Glu99, Thr100, Lys101, Leu102, Thr115, Tyr118) or α-helices (Ala16, Lys17, Asn20, Ala21, Met22, Lys23, Asp24, Ala25, Ser27, Lys31, Ile33, Asp123, Glu128, Gly134, Lys137, Gly138, Glu139, Ile145, Glu146, Tyr156). Uncertainties in the exchange parameters were estimated via a Monte Carlo approach, in which 100 synthetic datasets were generated using the exchange parameters obtained by the global fits described above, along with the experimental error in the rates (σ). Data fitting was repeated for all datasets. Errors quoted in the paper are standard deviations in fitted exchange parameters that were obtained in this procedure.

Ligand-observed ^1^H relaxation-dispersion experiments were performed as described in [34] to determine the binding kinetics of the ligand (-)-epicatechin on the 600 MHz spectrometer at 25 °C. Samples of Ara h 8.0101 at different concentrations (0, 40, 80, 120, 160, 200, and 240 µM) were prepared in 20 mM sodium phosphate buffer (pH 6.9), 9% D_2_O, and 2 mM (-)-epicatechin. The length of the CPMG element (T_relax_) was set to 60 ms, and the numbers of n cycles were set to 4, 8, 16, 24, 32, 40, 48, 56, 64, 72, 80, 88, 96, 104, 112, and 120, which resulted in ν_CPMG_ values of 66.7, 133.3, 266.7, 400.0, 533.3, 666.7, 800.0, 933.3, 1066.7, 1200.0, 1333.3, 1466.7, 1600.0, 1733.3, 1866.7, and 2000.0 Hz, respectively. Additionally, repeat experiments at 400.0 Hz, 933.3 Hz, and 1600.0 Hz were recorded. The length of the transverse relaxation filter (T_filt_) was set to 60 ms, and the number of transients was adjusted between 48 and 512 for the uncomplexed ligand to the highest ligand saturation. 

The TopSpin program (4.1.1) was used for the processing and analysis of the data. Baseline correction was performed between 5.5 ppm and 7.5 ppm (covering the ^1^H resonances of protons ^1^H-6/8 in (-)-epicatechin that were used for analysis). In each spectrum, the peaks were picked, and the absolute intensity at ν_CPMG_ = 0 (I_0_), and any given ν_CPMG_ (I_CPMG_), was determined. The conversion into effective relaxation rates and the fitting of the data to extract the off-rate of the ligand molecule (k_off_) were conducted as described [34].

### 4.3. NMR Titration Experiments

The titration of Ara h 8.0101 with (-)-epicatechin was performed by the stepwise addition of the (-)-epicatechin (90 mM stock solution in dimethyl sulfoxide (DMSO)) to 450 µL of a 0.2 mM ^15^N-labeled protein sample at 25 °C. The molar ratios of the Ara h 8.0101:(-)-epicatechin were 1:30, 1:28, 1:26, 1:24, 1:22, 1:20, 1:18, 1:16, 1:14, 1:12, 1:10, 1:8, 1:6, 1:4, 1:2, 1:1, and 2:1. The final concentration of dimethyl sulfoxide was below 6.67% (v/v). For each ratio, a ^1^H-^15^N-HSQC spectrum was recorded on the 700 MHz spectrometer. To identify amino acid residues that are involved in (-)-epicatechin binding, the combined amide chemical-shift perturbation was calculated as Δδ_obs_ = [((Δδ_H_)^2^ + (Δδ_N_/5)^2^)/2]^1/2^, where Δδ_H_ and Δδ_N_ are the chemical-shift changes observed for ^1^H and ^15^N, respectively [47]. A ^1^H-^15^N-HSQC spectrum was recorded by adding the maximum amount of DMSO (30 µL) to a 0.2 mM ^15^N-Ara h 8.0101 sample to evaluate the effect of DMSO on the observed chemical shifts (Δδ_DMSO_). The observed shift differences (Δδ_obs_) were then corrected for the DMSO, assuming a linear relation between the DMSO concentration and Δδ_DMSO_. A threshold value of 0.1 ppm was used to identify amino acid residues, which are affected by (-)-epicatechin binding. Residues exceeding this threshold were further used for determining the dissociation constant (K_D_) by the nonlinear least-squares fitting [33] of the Δδ_obs_, with an in-house written MATLAB script, assuming the binding of one equivalent of (-)-epicatechin per Ara h 8.0101 molecule, in agreement with the X-ray crystal structure [30].

### 4.4. In Silico Methods

For the structural comparison of the thirteen selected PR-10 allergens, we used NMR solution structures (Mal d 1.0101 (PDB: 5MMU), Pru p 1.0101 (PDB: 6Z98), and Cor a 1.040x (PDB: 6Y3H, 6Y3I, 6Y3K, 6Y3L)), or X-ray structures (Bet v 1.0101 (PDB: 4A88) and Ara h 8.0101 (PDB: 4M9B)) and CS-Rosetta structure models [48,49,50], based on the available backbone amide and side-chain Cα/Cβ NMR resonance assignments for Act c 8.0101 and Act d 8.0101 [41], and for Pru d 1.0101, Pru p 1.0201, and Pru p 1.0301 (unpublished). To classify and differentiate the PR-10 allergens based on their electrostatic and hydrophobic surface profiles, we generated images of the surfaces of all the PR-10 allergens, and we color-coded them based on the YRB highlighting scheme by using PyMol. The YRB highlighting scheme allows for the capture of both the hydrophobic and electrostatic properties of the respective protein surfaces [25]. This was performed by modifying the YRB script published by Hagemans et al. to not only be able to differentiate between the charged and hydrophobic regions, but also to include the H-bond donor/acceptor capacity in our coloring scheme. The clustering of these images was performed by using the image module implemented in Keras (Chollet, F., and others (2015). Keras. GitHub. Retrieved from https://github.com/fchollet/keras, accessed on 30 October 2021). The images were then preprocessed and converted into an array consisting of the features extracted during this process. We then applied k-means clustering and defined 3 clusters [51].

## Figures and Tables

**Figure 1 ijms-23-08252-f001:**
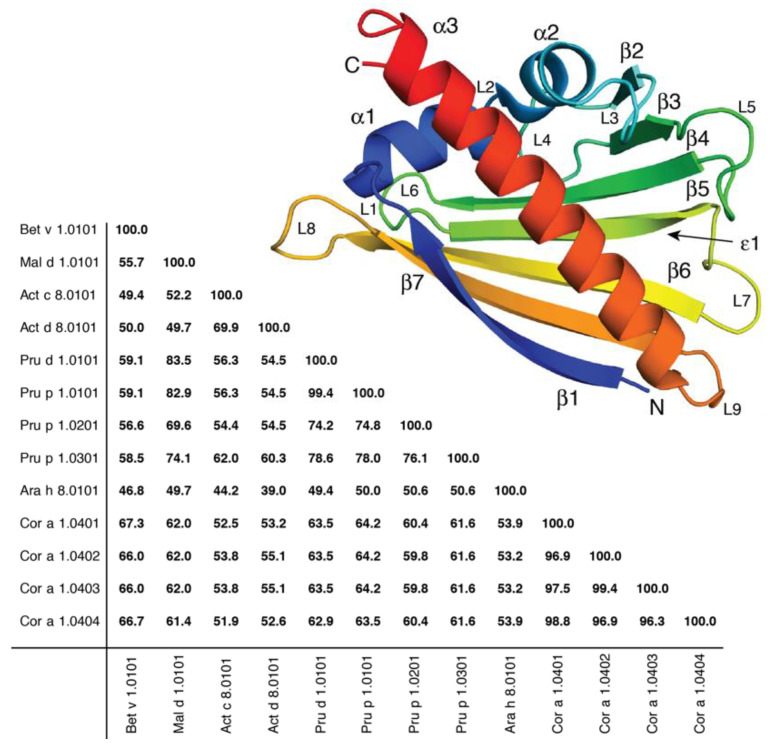
Sequence-identity matrix of thirteen selected PR-10 allergens from birch pollen (*Betula verrucosa*, Bet v 1), apple (*Malus domestica*, Mal d 1), golden kiwi fruit (*Actinidia chinensis*, Act c 8), green kiwi fruit (*Actinidia deliciosa*, Act d 8), plum (*Prunus domestica*, Pru d 1), peach (*Prunus persica*, Pru p 1), peanut (*Arachis hypogaea*, Ara h 8), and hazelnut (*Corylus avellana*, Cor a 1). The structure of Bet v 1 (PDB: 4A88, isoform Bet v 1.0101) is displayed above. Secondary-structure elements and the nine connecting loops (L1–L9) are labeled, and N- and C-termini are indicated. The main entrance (ε1) to the inner cavity of the protein is indicated by an arrow.

**Figure 2 ijms-23-08252-f002:**
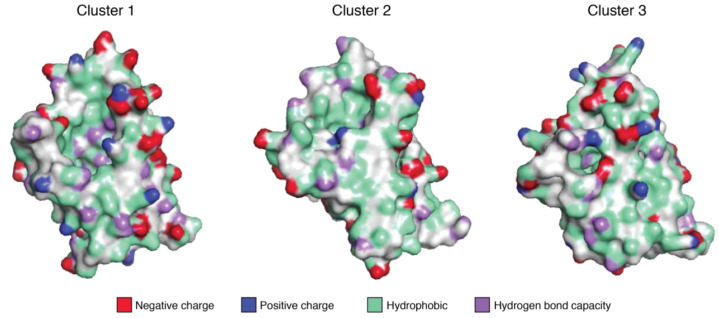
Clustering based on the charge profiles and hydrogen-bond donor/acceptor capacities of the thirteen PR-10 allergens. Cluster 1 comprises the proteins Cor a 1.0401, Cor a 1.0404, Mal d 1.0101, and Pru p 1.0101; Cluster 2 comprises Ara h 8.0101, Act d 8.0101, Bet v 1.0101, Cor a 1.0402, Cor a 1.0403, Pru d 1.0101, Pru p 1.0201, and Pru p 1.0301; Cluster 3 only contains Act c 8.0101. The surfaces are color-coded following the YRB highlighting scheme of Hagemans et al. [25]. The color-coding of positively and negatively charged residues, hydrophobic residues, and residues with hydrogen-bond capacities is specified. All three clusters are shown in the same orientation, with the C-terminal α-helix facing to the front, with its C-terminus pointing down.

**Figure 3 ijms-23-08252-f003:**
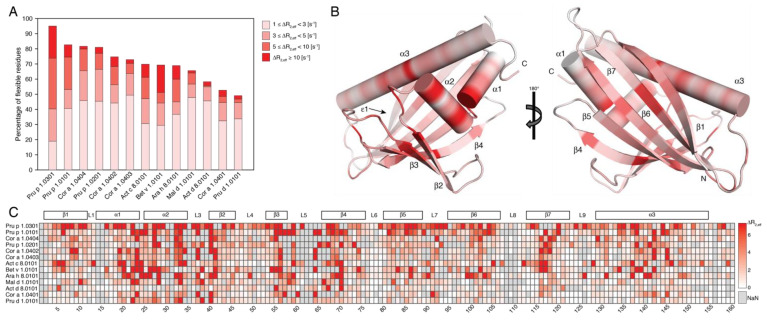
Structural flexibility of thirteen selected PR–10 allergens. (**A**) Bar plot showing the percentages of residues with backbone amide ^15^N relaxation-dispersion amplitudes (∆R_2,eff_) exceeding 1 s^–1^ at 700 MHz. Bars are color-coded according to ranges of ∆R_2,eff_ values. (**B**) Residue-specific mean values of ∆R_2,eff_ over all thirteen PR–10 proteins are plotted on the structure of Bet v 1.0101 (PDB: 4A88), color-coded from rigid (white) to flexible (red). (**C**) Heat map comparing backbone amide ^15^N RD amplitudes vs. residue numbers, colored according to the shown gradient. Proteins are ordered as in (**A**). Grey squares indicate sequence positions for which experimental data are not available (*i*) due to gaps in a specific PR–10 protein sequence, (*ii*) because of the resonance overlap of backbone amides, (*iii*) for prolines, or (*iv*) in cases where resonance assignments are not available.

**Figure 4 ijms-23-08252-f004:**
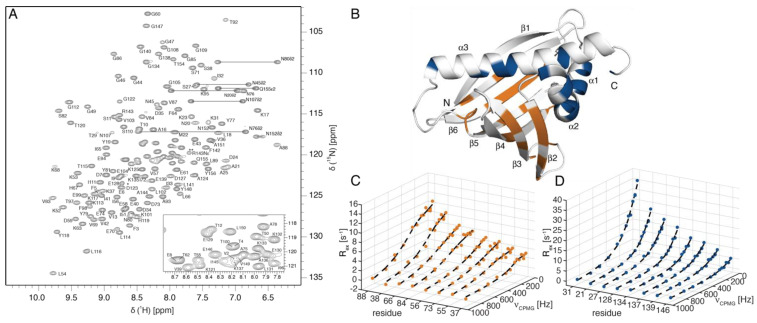
Backbone amide NMR resonance assignments and relaxation-dispersion data for Ara h 8.0101. (**A**) The 700 MHz ^1^H–^15^N–HSQC spectrum of Ara h 8.0101 (0.5 mM) in 20 mM sodium phosphate (pH: 6.9), supplemented with 10% D_2_O, at 25 °C. Assignments are shown using single–letter codes. Horizontal lines indicate asparagine and glutamine NH_2_ side–chain resonances. The resonance labeled by one asterisk indicates a residue below the intensity cutoff, the signal labeled by two asterisks indicates the aliased signal of the single arginine sidechain ε–NH. (**B**) Structure of Ara h 8.0101 (PDB: 4MA6) with backbone amides that were used in global fits of the ^15^N relaxation-dispersion data in the β-sheet and three α-helices, highlighted in orange and blue, respectively. (**C**,**D**) Backbone amide ^15^N relaxation-dispersion data for eight representative amino acid residues in the β-sheet (**C**) and in helices α1–α3 (**D**). Exchange contributions (R_ex_), which are calculated as R_2,eff_ – R_2,eff_ (ν_CPMG_ = ∞), are shown, along with the best fit-lines, which were obtained separately from the global fits of all the residues in (**C**,**D**), respectively.

**Figure 5 ijms-23-08252-f005:**
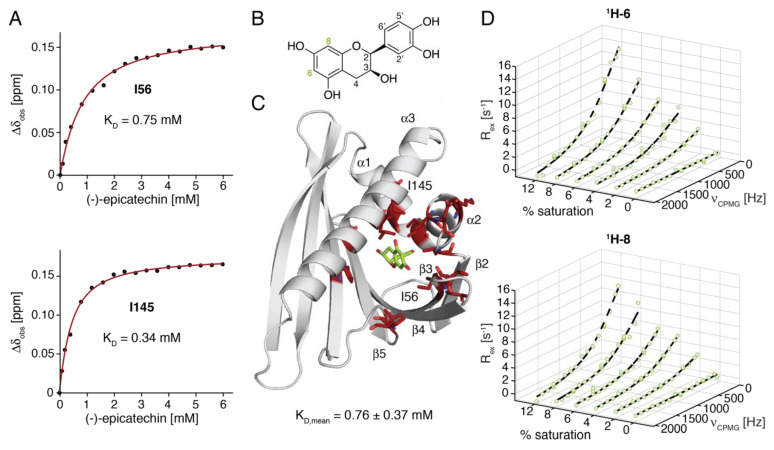
NMR spectroscopic analysis of binding of (–)–epicatechin to Ara h 8.0101. (**A**) Binding curves for two representative amino acid residues (Ile56 in strand β3, and Ile145 in helix α3). Site-specific K_D_ values were derived from ligand-induced ^1^H and ^15^N chemical-shift perturbations (Δδ_obs_) by nonlinear least-squares fitting [33]. (**B**) Structure and numbering scheme of (–)–epicatechin. (**C**) X-ray structure of Ara h 8.0101 in complex with (–)–epicatechin (PDB: 4MA6). Amino acid residues in red show backbone amides with Δδ_obs_ > 0.1 ppm upon (-)-epicatechin binding that were used to determine mean value and standard deviation of K_D_. (**D**) Ligand-observed ^1^H–CPMG relaxation-dispersion profiles, recorded at 600 MHz, for (-)-epicatechin binding to Ara h 8.0101. Data are shown for protons ^1^H–6/8 (indicated in green in panel (**B**)), and for increasing saturation (0–12%) of the ligand by protein. Solid lines represent the global fit to the data, yielding the off-rate (k_off_) of the (-)-epicatechin from the complex.

## Data Availability

Not applicable.

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
