# Peer review of "The Structural Flexibility of PR-10 Food Allergens"

_ijms, 2022, doi:10.3390/ijms23158252_

Round 1

Reviewer 1 Report

This paper of Martin Tollinger et al. deals with the structural flexibility of PR-10 food allergens. The applied technique is NMR relaxation dispersion spectroscopy. This method allows investigating sub-global conformational transitions on the millisecond time scale. This choice makes this work important since this low frequency motional domain can play important role in the mechanism of allergic reactions by emitting bound low-molecular-weight ligand from the internal cavity of protein. The flexibility of the investigated PR-10 allergens is well characterized by the backbone amide 15N relaxation dispersion profiles. Important result of this work pointing out that that all allergens have inherently flexible protein backbones in solution, yet the extent of structural flexibility appears to be strongly protein specific.

Questions:

- How this protein specificity is related to the structure of the released low molecular weight ligand? - - What is the most important for the allergic reactions: the properties of the emitting protein or the emitted ligand molecules?

Since this paper discusses important questions and applies proper experimental work, and the conclusions are also reasonable, it can be published after minor revision.

Author Response

Thank you for reviewing our manuscript and your valuable recommendations.

  1. How this protein specificity is related to the structure of the released low molecular weight ligand? The specificity of PR-10 proteins is generally low, with few exceptions. This is now outlined in lines 82-84 of the manuscript. 
  2. What is the most important for the allergic reactions: the properties of the emitting protein or the emitted ligand molecules? The allergenic behavior is mostly determined by the protein and not by the ligand. Whether and how ligand binding to PR-10 allergens is related to their immunologic properties is currently under debate, as stated in lines 85/86 of the manuscript. References are included.

Reviewer 2 Report

This interesting manuscript shows a group of structurally and evolutionarily related proteins (pathogenesis-related, PR) show a rather unusually high level of flexibility. I am not expert in these proteins, but the introduction tells us that they are allergenic. I think a conclusion made is there is not a relationship of how allergenic the protein is to how flexible/rigid the individual proteins are. Nevertheless, I find the dynamics quite fascinating and this manuscript adds to the body of knowledge in protein science and specifically on protein dynamics. The dynamics may be selected by nature for the protein's capacity to interact with a range of natural ligands for their function as defense proteins. The manuscript characterizes the millisec timescale motion using dispersion NMR experiments. Technically, the work is very well executed. The dynamics points to additional conformations involving both beta sheet and helices. The beta sheet residues undergo the largest excursion (on a slightly slower timescale), but the data does not point to an unfolding of the protein, rather additional state(s). The actual kinetics of the sheet and helices differ - very interesting. I feel there may be a common underlying "cause" but that remains to be discovered. I have very little to fault about the manuscript; it is well written and very clear. A few minor suggestions:

line 62 state what loop the GXGGXG motif is in (you do much later but would be helpful here.

line 85 ...allergenic behaviour (include references to this sentence?) 

line 86-88 when you say related to ligand binding, do you mean affinity, specificity? I think qualify.

Throughout write KD and not Kd

Supplementary Figure 2; what was the reasoning for the overlays in B; I don't think they agree with those you have clustered in Figure 2, so I am not sure on the rationale.

Author Response

Thank you for reviewing our manuscript and your valuable recommendations.

  1. line 62 state what loop the GXGGXG motif is in (you do much later but would be helpful here: We now include this information already in line 61 of the manuscript.
  2. line 85 ...allergenic behaviour (include references to this sentence?): We included a reference to a recent review by Morris et al., which is describing this issue (line 89)
  3. line 86-88 when you say related to ligand binding, do you mean affinity, specificity?: We mean affinity rather than specificiy, and modified this sentence accordingly (line 91)
  4. We now use KD exclusively
  5. Supplementary Figure 2; what was the reasoning for the overlays in B: This figure is independent from Figure 2. It merely intends to overlay allergens from the same (or similar) food sources.